# The Development and Synthesis of a CdZnS @Metal–Organic Framework ZIF-8 for the Highly Efficient Photocatalytic Degradation of Organic Dyes

**DOI:** 10.3390/molecules28237904

**Published:** 2023-12-02

**Authors:** Liu Hong, Jiaming Cao, Wenlong Zhang, Tao Jiang, Guohao Pan, Yun Wu

**Affiliations:** School of Energy Materials and Chemical Engineering, Hefei University, Hefei 230601, China; hl18324702376@163.com (L.H.); cjm15155934415@163.com (J.C.); zwnl1628069773@163.com (W.Z.); m158554288245@163.com (T.J.); guohaopan@163.com (G.P.)

**Keywords:** CdZnS, ZIF-8, methylene blue, dye wastewater, photocatalytic reaction

## Abstract

The development of photocatalysts for organic degradation is a hot research topic. In this study, CdZnS was selected as the carrier, and ZIF-8 was combined with it to explore the photocatalytic performance of the composite. In addition, the compound material, CdZnS@ZIF-8, was used as a photocatalyst for the decomposition of methylene blue dye, and the performance of pure CdZnS and pure ZIF-8 was compared. The photocatalytic efficiency of CdZnS@ZIF-8 was significantly higher than that of the other two. In the experimental reaction, the amount of catalyst was 0.04 g, the pH value was 7, the initial concentration of methylene blue aqueous solution was 20 mg/L, and the degradation of methylene blue in 50 mL aqueous solution could reach 99.5% under visible light irradiation for 90 min, showing excellent photocatalytic efficiency in the visible light range. It demonstrated excellent photocatalytic function in the visible light region, and the electron transfer phenomenon at the interface occurred in the het-junction and the separation of the photo-generating electron–hole as an electron acceptor of ZIF-8 further promoted the photocatalytic effect.

## 1. Introduction

The effective treatment of dye wastewater has become an important problem to be solved urgently in the field of water environmental pollution control. Most organic dyes, as major water pollutants, are extremely difficult to treat due to their stability, carcinogenicity, and low biodegradability in aerobic digestion [1,2,3,4,5]. The release of dye-containing wastewater into the environment causes serious harm to humans and other life forms on Earth. Therefore, it is very important to reduce the concentration of pollutants before they are released into the environment.

Photocatalysis is a process that uses light energy and catalysts to promote chemical reactions. In a photocatalytic reaction, light energy is absorbed and converted into excited electrons, which can react with oxygen, water, or other molecules in solution to produce highly reactive free radicals or oxidants that trigger chemical reactions. In the process of the photocatalytic degradation of dyes, the photocatalyst is usually a semiconductor material such as titanium dioxide (TiO_2_) or zinc oxide (ZnO) [6]. These photocatalysts can absorb visible or ultraviolet light and produce excited electrons and hole pairs under light. Photocatalytic oxidation has been used to study the degradation of dye wastewater due to its convenient operation, low cost, and wide applicability [7,8,9,10,11]. In this study, the authors mainly studied the photocatalytic decomposition of dyes.

As a branch of MOF materials, zeolite imidazole frameworks (ZIFs) are composed in the same way as MOFs materials, mainly consisting of Zn^2+^ or Co^2+^ as metal chemical coordination sites, and imidazole or imidazole derivatives as organic ligands, which are formed through chemical bonds [12,13,14,15,16,17,18,19,20]. Compared with traditional zeolite materials, the metal ions contained in ZIF materials are replaced with silicon and aluminum elements in zeolite, forming zeolite-like structural materials with imidazole as the skeleton. ZIFs are a new porous structural material that combine the advantages of MOFs with the characteristics of zeolite material [21,22,23,24,25]. ZIFs generally have higher thermal and chemical stability than other MOFs, which lays a good foundation for the application of ZIFs in catalysis, gas storage, and separation [26,27]. The synthesis of ZIF-8 is typically carried out using solvothermal or solvent evaporation methods, where metal ions and organic ligands come together to form a crystal structure under specific conditions. The structural characteristics of ZIF-8 include a highly organized pore structure, the ability to adjust pore size, a large surface area, and excellent thermal stability.

Zinc cadmium sulfide (Cd_0.5_Zn_0.5_S) is a solid solution of CdS and ZnS, hereinafter referred to as CdZnS [28]. With an appropriate band gap (2.4 eV), the material has good visible light absorption capacity and good transmission in the visible light region [29]. Furthermore, the conduction band (CB) side potential of CdZnS is more negative than the reduction potential of H_2_O/H^2^; therefore, it is considered a promising photocatalyst. However, CdZnS also has shortcomings such as the rapid recombination of photogenerated carriers and photo corrosion during the photocatalysis process, and its photocatalytic activity can be improved by doping metal elements or combining them with other semiconductor materials [30].

In this study, the authors constructed a composite material which formed a heterojunction structure between CdZnS particles and the organometallic skeleton ZIF-8, which could effectively reduce the electron–hole recombination rate and promote its absorption of visible light, so that the composite photocatalyst had a stronger photocatalytic performance. In addition, it could prevent the secondary agglomeration of the material and maintain the superior adsorption properties of the ZIF-8 channel [15,31].

## 2. Results and Discussion

### 2.1. Characterization Analysis

Powder X-ray diffraction (XRD), scanning electron microscopy (SEM), X-ray photoelectron spectroscopy (XPS), N2 adsorption–desorption, and UV–Vis–NIR diffuse reflection spectroscopy (UV-VIS-NIR DRS) were used to characterize the prepared materials.

Figure 1 shows an XRD comparison of standard CdZnS, prepared CdZnS, standard ZIF-8, and the composite CdZnS@ZIF-8. All characteristic peaks of the prepared CdZnS correspond to the standard CdZnS characteristic peaks, indicating that the purity of the prepared CdZnS meets the experimental requirements [32]. The main diffraction peak shows a broadening phenomenon, which indicates that the grain is finer. The standard XRD pattern of the composite material CdZnS@ZIF-8 is in good agreement with that of ZIF-8, and most of the characteristic peaks correspond to ZIF-8 [33]. It is speculated that ZIF-8 accounts for a large proportion of the composite material, but there are still some mixed peaks, which may be residual solvent molecules.

The purity, phase composition, and crystallinity of the resulting products were investigated by powder X-ray diffraction (XRD) [34]. Figure 2a shows a single particle of ZIF-8, whose average grain size is between 200 nm and 300 nm according to statistical analysis. The particles are dodecahedral, with very uniform particle size and good inter-particle dispersion [31]. Figure 2b shows the CdZnS powder prepared in the experiment, most of which are chrysanthemum-like. The particle size ranges from 1 um to 2 um, and the particle size distribution range is wide. In Figure 2c,d, the scarce CdZnS chrysanthemum-like particles in the solution have absorbed many layers of ZIF-8 unevenly, and CdZnS is completely encased in cubes, forming a rigid organometal skeleton shell.

To study the chemical composition and elemental valence states on the surface of photocatalysts, X-ray photoelectron spectroscopy (XPS) analysis was performed. XPS measurement scans revealed the presence of O, N, C, Cd, Zn, and S elements in the composite sample, as shown in Appendix A. Appendix A shows the C1s spectrum of CdZnS@ZIF-8, with three peaks at 288.5 eV, 285.5 eV, and 284.5 eV, corresponding to C-C/C=C, C-N(2-methylimidazole), and O-C-O, respectively, as shown in Appendix A. The two peaks at 530.95 eV and 531.7 eV correspond to oxygen vacancies and O^2−^ in CdZnS on the surface of the material, respectively. Appendix A depicts the XPS spectrum of Zn 2p. Two characteristic peaks appear at 1044.8 eV and 1021.8 eV, corresponding to Zn 2p 1/2 and Zn 2p 3/2, respectively, indicating that Zn exists as a divalent zinc ion (Zn^2+^) [35].

The light absorption ability and range of a photocatalyst are critical to photocatalysis because photon energy is the driving force for the generation of charges [6]. Using ultraviolet–visible spectra (UV–Vis), the light absorption properties of ZIF-8, CdZnS, and CdZnS@ZIF-8 were studied. As shown in Appendix A, ZIF-8 has weak absorption in the ultraviolet region, while CdZnS has wide absorption in the ultraviolet region and visible light region. After the combination of ZIF-8 and CdZnS, compared with ZIF-8, the light absorption capacity of the composite CdZnS@ZIF-8 is significantly enhanced in both the ultraviolet region and the visible region. On the one hand, the two materials have different absorption capacities, and the overall light absorption capacity of the composite is improved. However, composites have large specific surface areas and typical multistage structures (as explained in the following BET representation test), the light can be fully and effectively diffused in its interior, and the utilization rate of CdZnS@ZIF-8 is improved.

Thermogravimetric analysis was used to study the change in catalyst mass with the increase in temperature program, to measure the thermal stability of the skeleton structure. It can be seen from Appendix A that ZIF-8 has two obvious weightlessness processes between 30 and 800 °C. Between 100 °C and 200 °C, the weight loss is about 10%. The weight loss in this process is caused by the evaporation of trace water molecules in the mesopore of ZIF-8 and the removal of part of the organic ligand, which is also the process of the formation of its defects [36]. Above 600 °C, ZIF-8 showed large weight loss, indicating that the structure of ZIF-8 had begun to be destroyed after 600 °C. CdZnS had good thermal stability, with a mass loss of 6.5% after 800 °C. Compared with pure ZIF-8, the composite CdZnS@ZIF-8 maintained a good structure at room temperature up to 400 °C, and exhibited roughly the same condition as ZIF-8 at 600 °C.

The BET surface area is an important property of porous materials, which is related to the pore structure and pore size of porous materials. A larger BET surface area means that more active surfaces are available for reactions to occur, providing more reaction sites and adsorption sites. This can increase the activity of the photocatalytic reaction because more reactants can make contact with the photocatalyst and react. The specific surface areas and opening distribution curves of pure ZIF-8, CdZnS, and CdZnS@ZIF-8 were measured by N2 adsorption and dissolution experiments. As shown in Figure 3a,e, the adsorption–desorption isotherm of pure ZIF-8 and the composite material CdZnS@ZIF-8 has a sharp adsorption inflection point in the low-pressure range of 0–0.1, which is a typical type I curve. There is a well-shaped platform at high relative pressure, which indicates microporosity [37]. CdZnS, as shown in Figure 3c, presents a type III isotherm and is a microporous solid material. The interaction between the adsorbed material and the adsorbed gas is relatively weak, and the adsorbed molecules are concentrated around the most attractive parts of the surface. At the same time, it can be seen from Figure 3e that the hysteresis curve is of H4 type, indicating that in addition to the microporous structure, the synthesized composite material has some mesoporous structures similar to those generated by the layered structure [38,39]. In addition, it can be seen from Appendix A that the BET specific surface area of the composite material is 1151.684 m^2^g^−1^, which is slightly smaller than that of ZIF-8 (1836.352 m^2^g^−1^) and much larger than that of CdZnS (12.662 m^2^g^−1^). It may be that ZIF-8 wraps around CdZnS. By analyzing the Barrett–Joyner–Halenda (BJH) pore size distribution of the CdZnS@ZIF-8 composite (Figure 3f), the composite has a three-stage pore structure with micropores, mesoporous pores, and macropores. The results show that CdZnS@ZIF-8 composites with multistage pore structures can reflect and absorb incident light many times on the one hand, and accelerate the transfer efficiency and mass diffusion in the photocatalytic process on the other hand.

### 2.2. Photocatalytic Performance Evaluation

In a typical scheme, the 50 mL dye solution with a mass concentration of 10 mg/L was mixed with 20 mg of CdZnS@ZIF-8 composite material or ZIF-8 or CdZnS prepared by different methods. After ensuring that the solution was not exposed to light, it was oscillated for 30 min. A 300 W xenon lamp was used to simulate the irradiation of sunlight at 30 cm from the reaction solution to be measured (irradiance: 735 W/m^2^). A UV–VIS spectrophotometer was used to determine the mass concentration of dye solution at different light times, and the degradation rate or adsorption rate (D, %) was calculated according to Equation (1):D/% = [(ρ_0_ − ρ_e_)/p_0_] × 100 (1)
where ρ_0_ and ρ_e_ are the start of the dye and the equilibrium mass concentration, respectively, for a certain illumination time, mg/L.

To obtain CdZnS@ZIF-8 composite material with good catalytic performance, composites with different mass ratios were prepared, and ZIF-8 and CdZnS before synthesis were used for the photocatalytic degradation of methylene blue. In the experiment, the initial concentration of methylene blue was specified as 30 mg/L and the dosage of photocatalytic degradation material was 50 mg. This did not change the pH of the solution. Figure 4a shows the photocatalytic degradation of methylene blue with different mass ratios. It can be seen from the figure that except for 1% material, the removal rate curve of the remaining materials increased greatly during 0–10 min of illumination, and then linearly tended to a stable rise process; the removal rate was proportional to time, while the removal rate of 1% showed a steady rise with the increase in time. After a reaction time of 90 min, the removal rate of 0.8% composite photocatalytic degradation of methylene blue reached 99.54%, while the removal rate of ZIF-8 was 96.32%, and the removal rate of CdZnS was only 63.73%. The experimental results showed that the photocatalytic degradation of methylene blue by 0.8% composite was not only improved by ZIF-8 and CdZnS, but also increased the degradation rate.

Figure 4b shows the effect of the photocatalytic degradation of methylene blue with different amounts of materials. As shown in Figure 4b, when the dosage was 10 mg to 40 mg, the removal rate showed an upward trend with the increase in the dosage of material. When the dosage of the material was 40 mg, the removal rate of methylene blue reached 99.55%. This is because when the amount of material is small, the photochemical particles generated by the light source are not fully converted into chemical energy, resulting in the light energy not being fully utilized, and an appropriate increase in the number of degradable materials can produce more active species and increase the efficiency of photocatalytic degradation. According to a previous review and the wider literature, it was found that ZIF-8 composite material itself contains metal active sites, and the mesoporous structure can meet the mass transfer process of macromolecular substances at the same time. By increasing the dosage of materials, the active sites are increased, which can improve the photocatalytic degradation efficiency of materials. Therefore, when adding photocatalytic materials, the amount of material added should be appropriately increased, and the amount of material added should be 40 mg.

After 30 min, the experiment entered the photocatalytic degradation stage (Figure 4c). When the light time was 0–10 min, the removal rate increased rapidly and showed a steep rise. Subsequently, the rising trend gradually stabilized, and the removal rate was proportional to time. This is due to the simultaneous catalysis and adsorption of composite material in methylene blue degradation solution in 0–10 min, resulting in a significant increase in the removal rate. Later, in the process of photocatalytic degradation, the adsorption effect of each sample is reduced, and photocatalytic degradation plays a key role. With the extension of reaction time, the number of active sites on the surface of the material is reduced, and the driving force of the degradation reaction is gradually reduced, resulting in the degradation rate gradually slowing down until the chemical reaction equilibrium is reached. The removal rate of samples with different concentrations of methylene blue solution is different after 50 min, showing a decreasing trend. This is because when the concentration of methylene blue solution is low, the dye can rapidly occupy on the active site of the degradation reaction of the material and increase the removal rate. When methylene blue solution is high in concentration, the dye needs to be diffused internally to the surface of the degradation material. However, the steric repulsion between solute molecules will slow down the adsorption process, resulting in a decrease in the removal rate.

Figure 4d shows the effect of the photocatalytic degradation of methylene blue under pH 7–9 conditions. It can be seen from Figure 4d that effective degradation can occur under near-neutral and alkaline conditions, indicating that the photocatalytic degradation of methylene blue by the synthesized material has a wide range of applications in alkaline conditions. When pH = 7, the removal rate of Rhodamine B reaches 99%, and the removal rate decreases with the increase in pH, possibly because a large amount of OH^−^ will occupy the active site of the material when the pH is too high [40]. The pore sizes of materials with mesoporous structures make it easier for excess H^+^ and OH^−^ to enter the pores of the material and occupy active sites, resulting in a decrease in removal rate.

It can be seen from Figure 4e that with the increase in the number of experiment cycles, the removal rate showed a downward trend, because the desorption process of the composite material for methylene blue was not complete. The removal rate decreased steadily after the fourth cycle, which may be because the degradation process reached chemical equilibrium. After five cycles, the removal rate of CdZnS@ZIF-8 for methylene blue could still reach 83.25%. The results show that the composite still had a high photocatalytic activity after repeated recycling, showing good reusability and stability.

In order to evaluate the selectivity of CdZnS@ZIF-8 for methylene blue and further investigate its photocatalytic degradation of organic dyes, the following experiments were designed. Under the condition that 40 mg CdZnS@ZIF-8 was added and the initial pH of the solution was set to 7, we selected different dyes with a concentration of 20 mg/L, including methyl orange (MO), Rhodamine B (RB), methylene blue (MB), and reactive brilliant red X-3B, for photocatalytic degradation tests. The experimental results are shown in Figure 4f.

After 30 min dark reaction and 90 min light reaction, the photocatalytic degradation rates of CdZnS@ZIF-8 for methyl orange, Rhodamine B, methylene blue, and reactive brilliant red X-3B were 60.56%, 81.32%, 99.55%, and 86.47%, respectively. These results indicated that CdZnS@ZIF-8 had high removal efficiency and showed good catalytic degradation performance for these four different dyes. In particular, the removal rate of methylene blue was the highest, reaching 99.55%. Therefore, we can conclude that CdZnS@ZIF-8 is not selective to methylene blue and shows excellent photocatalytic degradation performance in the treatment of dye wastewater.

### 2.3. CdZnS@ZIF-8 Photocatalytic Mechanism of Materials

Theoretically, the photocatalytic reaction of semiconductor materials can be divided into three stages: first, the generation of electron–hole pairs, then the separation of electron–hole pairs, and finally the transfer to the catalyst surface. On this basis, the photogenerated electrons and holes react on different surface adsorption media. Under photon irradiation with energy higher than Eg, CdZnS can be excited to produce photogenic carriers, but the carriers are easy to aggregate. In addition, oxidation and reduction reactions occur in the positive and inverse bands. More reactions occur, producing ·OH [41,42]. The essence is to use ·OH to oxidize organic pollutants in water and produce CO_2_ and H_2_O [43].

As shown in Figure 5, when CdZnS forms a heterojunction with ZIF-8, if the conductivity of the two is opposite, the electron–hole transfer between the two will continue until the Fermi levels of the two are equal [44]. Among them, the interface between ZIF-8 and CdZnS forms a space charge region. One side of ZIF-8 is negative, while the other side of CdZnS is positive; the positive and negative charge regions form an internal electric field to generate a heterojunction [45]. The photogenerated electron–hole pair in the CdZnS conductive band is generated in CdZnS; with the self-generated electric field, photogenic holes are transferred from CdZnS to ZIF-8, and photogenic electrons remain in CdZnS. The self-generated electric field can effectively inhibit electron–hole recombination, making the electron–holes participate in the photocatalytic reaction in CdZnS. Thus, the photocatalytic reaction rate can be improved.

## 3. Experimental Section

### 3.1. Preparation of ZIF-8

At room temperature, 2-methylimidazole was synthesized using organic bridging ligands and inorganic metal nodes provided by zinc acetate in a methanol solution. To initiate the synthesis, 3.2836 g of 2-methylimidazole and 2.195 g of zinc acetate were individually weighed and dissolved in a beaker containing 70 mL of methanol solution. The resulting mixture was thoroughly mixed and sealed. The mixture was then stirred at room temperature for 24 h. Subsequently, the mixture was subjected to centrifugation at 7000 RPM for 5 min, and the separated solids were washed three times with 50 mL of methanol. Finally, the washed solids were dried in an oven at 60 °C for 24 h.

### 3.2. Preparation of CdZnS

In a beaker containing 20 mL of deionized water, 0.18 g (1.0 mmol) of CdCl_2_•2.5H_2_O, 0.292 g (1.0 mmol) of zinc acetate, and 0.15 g (2.0 mmol) of TAA were added. The mixed solution was stirred for 1 h and then transferred to a hydrothermal reactor. The hydrothermal reaction was carried out at 160 °C for 12 h. After completion, the reaction mixture was cooled to room temperature, filtered, and the resulting Cd_0.5_Zn_0.5_S precipitate was washed several times with anhydrous ethanol and deionized water. The washed precipitate was then dried at 70 °C for 12 h and set aside for further use.

### 3.3. Synthesis of CdZnS@ZIF-8

First, 0.012 g (0.0001 mol) of CdZnS was accurately weighed and added to a 70 mL methanol solution for ultrasonic dispersion for 30 min. Subsequently, 3.284 g (0.04 mol) of 2-methylimidazole and 2.195 g (0.01 mol) of zinc acetate were added to another 70 mL methanol solution for mixing and stirring until dissolved. The two methanol solutions were then combined and allowed to react at room temperature for 24 h with continuous stirring.

After the reaction, the resulting precipitate was collected by centrifugation and washed three times with methanol solution. The washed precipitate was then dried in a drying oven at 60 °C for 12 h. Once dried, the obtained product was ground into a powder, transferred into a sample tube, and labeled accordingly. Subsequently, the sample underwent characterization testing and analysis.

## 4. Conclusions

In summary, in this study, CdZnS was selected as the carrier, ZIF-8 was combined with it, and a composite material with good photocatalytic performance was successfully prepared. This is a new structure of three-dimensional CdZnS particles combined with a metal–organic skeleton. A multistage porous composite with microporous, mesoporous, and microporous structures was constructed. The experimental results showed that the photocatalytic efficiency of CdZnS@ZIF-8 was much higher than that of pure CdZnS and pure ZIF-8. In the experimental reaction, the amount of catalyst was 0.04 g, the pH value was 7, the initial concentration of methylene blue aqueous solution was 20 mg/L, and the degradation of methylene blue in 50 mL aqueous solution reached 99.55% under visible light irradiation for 90 min, which demonstrates the excellent photocatalytic efficiency in the visible light range. The preparation method of the composite is simple, which not only broadens the absorption range of ZIF-8 to visible light, but also improves its photocatalytic activity, which provides feasibility for the application research of photocatalytic degradation of materials.

## Figures and Tables

**Figure 1 molecules-28-07904-f001:**
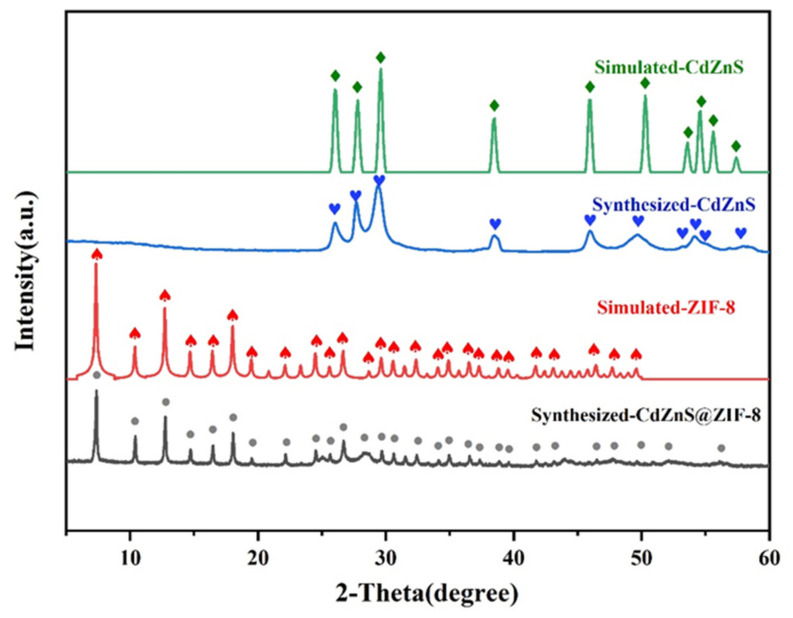
XRD patterns of simulated CdZnS, synthesized CdZnS, simulated ZIF-8, and synthesized CdZnS@ZIF-8.

**Figure 2 molecules-28-07904-f002:**
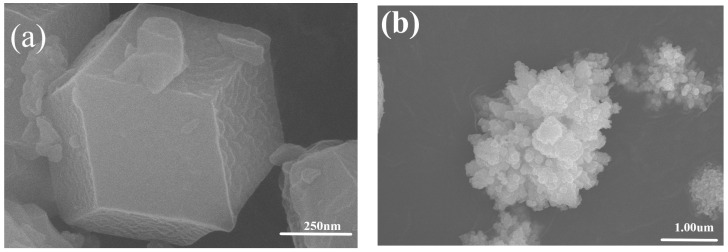
SEM images of (**a**) ZIF-8, (**b**) CdZnS, and (**c**,**d**) CdZnS@ZIF-8.

**Figure 3 molecules-28-07904-f003:**
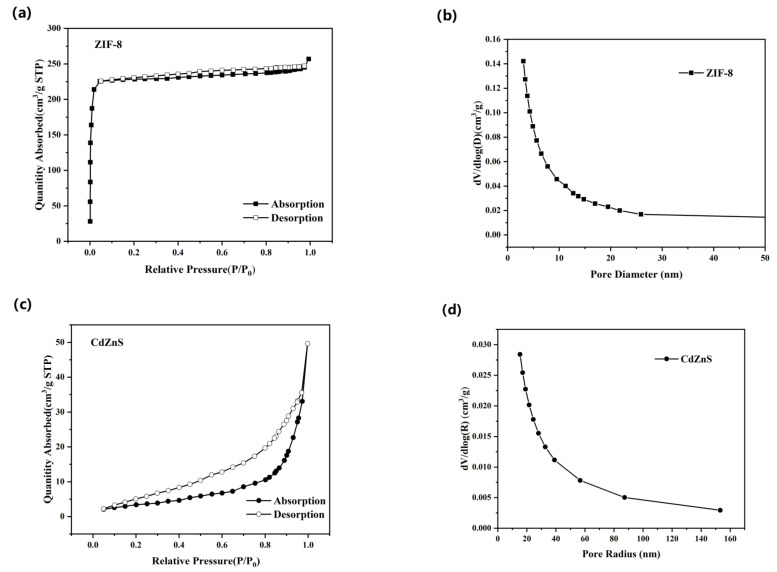
(**a**,**c**,**e**) N2 adsorption–desorption isotherms and corresponding pore size distribution curves (**b**,**d**,**f**) for pure ZIF-8, CdZnS, and CdZnS@ZIF-8 composites.

**Figure 4 molecules-28-07904-f004:**
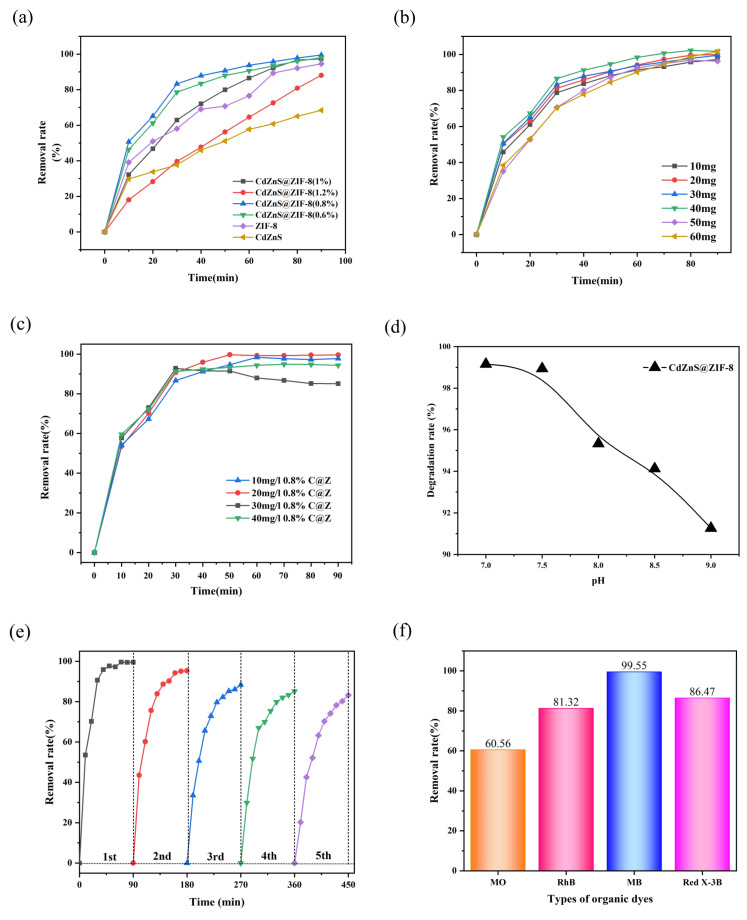
(**a**) Comparison of methylene blue degradation by photocatalysis with different mass ratios; (**b**) comparison of methylene blue degradation by photocatalysis with different dosages; (**c**) comparison of the photocatalytic degradation of methylene blue at different initial concentrations; (**d**) comparison of the photocatalytic degradation of methylene blue at different pH concentrations; (**e**) CdZnS@ZIF-8 cyclic stability of composites; (**f**) degradation rate of different kinds of dyes.

**Figure 5 molecules-28-07904-f005:**
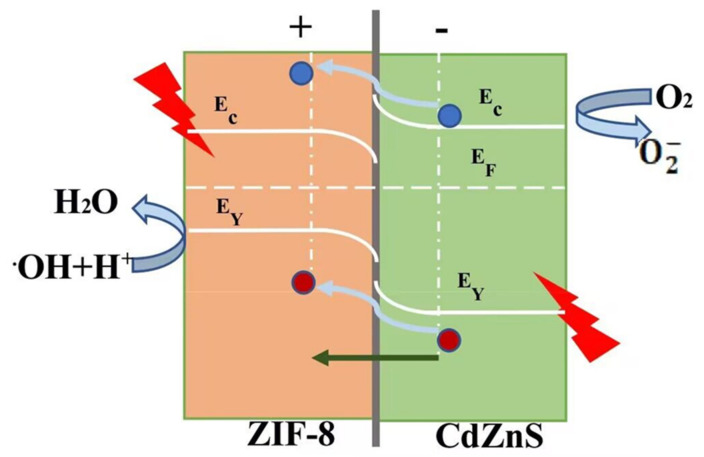
Schematic diagram of electron-hole transfer in CdZnS@ZIF-8 composites.

## Data Availability

The data presented in this study are available in the Appendix A.

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
