# Peer review of "The Development and Synthesis of a CdZnS @Metal–Organic Framework ZIF-8 for the Highly Efficient Photocatalytic Degradation of Organic Dyes"

_molecules, 2023, doi:10.3390/molecules28237904_

Round 1

Reviewer 1 Report

Comments and Suggestions for Authors

The manuscript entitled “Development and synthesis of CdZnS @Metal-Organic Framework ZIF-8 :Highly efficient photocatalytic degradation of organic dyes” is an interesting work on the development of a new composite for photocatalytic processes. The work is interesting but not well-organized, it lacks scientific content in some aspects and is not well-written. A complete review of grammar, semantics and structure is required. Some specific comments are below:

ABSTRACT

1) Line 56: “In this paper, we hope to construct”… Why “hope”? This is not really scientific. Also, never use “I”, “we” in scientific works. Check it throughout the manuscript.

MATERIALS AND METHODS

2) Line 66: There are numbers with 3 decimals and others with 4… Please, be uniform.

3) The same for other numbers throughout the manuscript.

4) Line 74: Caps look in Transfer?

5) Line 73-76: The preparation description seems a receipt. It should be described how it was done, not how must be done (receipt).

RESULTS AND DISCUSSION

6) Line 166: Again, these sentences are not well-written. It seems a receipt. It is written in the imperative form.

7) The manuscript should be carefully revised.

Comments on the Quality of English Language

A complete review of grammar, semantics and structure is required.

Author Response

Response to Reviewer 1 Comments

Point 1: Line 56: “In this paper, we hope to construct”… Why “hope”? This is not really scientific. Also, never use “I”, “we” in scientific works. Check it throughout the manuscript.

Response 1: The original text has been revised in line 63, and the "we" and" i" in the article have been scientifically modified and replaced.

Point 2:  Line 66: There are numbers with 3 decimals and others with 4… Please, be uniform.

Response 2: The numbers after the decimal point have been unified

Point 3: The same for other numbers throughout the manuscript.

Response 3: The numbers after the decimal point have been unified

Point 4: Line 74: Caps look in Transfer?

Response 4: Syntax error, has been fixed.

Point 5: Line 73-76: The preparation description seems a receipt. It should be described how it was done, not how must be done (receipt).

Response 5: The original expression is not appropriate enough, and the preparation process of the article has been comprehensively rewritten and modified.

Point 6: Line 166: Again, these sentences are not well-written. It seems a receipt. It is written in the imperative form.

Response 6: The original expression is not appropriate enough, so we have revised it

Reviewer 2 Report

Comments and Suggestions for Authors

The authors presented, “Development and synthesis of CdZnS @Metal-Organic Framework ZIF-8 Highly efficient photocatalytic degradation of organic dyes.” It is a topic of interest to the researchers in the related areas, but the paper needs very significant improvement and explanation before acceptance for publication.

1.      What is the leaching concentration of Cd and Zn ions in this photocatalytic system? What is the contribution of leaching ions on Methylene blue degradation?

2.      Besides Methylene blue, the catalytic performance in other organic pollutants should also be investigated.

3.      The innovation of the article should be clearly stated

4.      If possible, check the toxicity of the effluent or is determined the BOD5/COD value.

5.      If possible, TOC for different catalytic systems should be compared.

Author Response

Response to Reviewer 2 Comments

Point 1: What is the leaching concentration of Cd and Zn ions in this photocatalytic system? What is the contribution of leaching ions on Methylene blue degradation?

Response 1: Zn ion action: Zn ion can be used as an assistant in the photocatalyst to enhance the efficiency of the photocatalytic reaction. In the photocatalytic reaction, Zn ions can interact with the active site on the surface of the photocatalyst to improve the adsorption capacity of the photocatalyst and the separation efficiency of the photogenerated electron-hole pairs. Zn ion has good light absorption ability, can absorb visible light and produce excited electrons. In addition, Zn ions can also react with oxygen molecules in solution to produce reactive oxygen species, such as superoxide free radicals (O2•-), which further promote the degradation of methylene blue.

Cd ion action: Cd ion can play the role of photosensitizer in photocatalytic reactions. Cd ion has a narrow band gap and high light absorption capacity, which enables it to absorb ultraviolet light and produce excited state electrons. These excited electron-hole pairs can participate in the degradation reaction of methylene blue, thus accelerating the degradation rate of methylene blue. In addition, Cd ions can also react with oxygen molecules in the solution to produce reactive oxygen species, such as hydroxyl radicals (•OH), which further promote the degradation of methylene blue.

The mechanism of photocatalysis is also discussed in this paper.

Point 2:  Besides Methylene blue, the catalytic performance in other organic pollutants should also be investigated.

Response 2: In addition to methylene blue, the study of catalytic properties in other organic pollutants has been added, such as methyl orange, Rhodamine B, reactive brilliant red X-3B. It is found that the material also has a good ability to degrade these organic pollutants, and I have compared the performance of these results. We can see lines 237 to 251 of the manuscript.

Point 3: The innovation of the article should be clearly stated

Response 3: 1. A new structure of three-dimensional CdZnS particles combined with metal-organic skeleton was constructed

  1. CdZnS composite not only broadens the absorption range of ZIF-8 to visible light, but also improves its photocatalytic activity
  2. A multistage porous composite material with microporous, mesoporous, and microporous structure was constructed.

Point 4: If possible, check the toxicity of the effluent or is determined the BOD5/COD value.

Response 4: I'm sorry that the laboratory does not have the conditions for this test now.

Point 5: If possible, TOC for different catalytic systems should be compared.

Response 5: I'm sorry that the laboratory does not have the conditions for this test now.

Reviewer 3 Report

Comments and Suggestions for Authors

Reviewer comments

In this article author work on “ Development and synthesis of CdZnS @Metal-Organic Frame-2 work ZIF-8:Highly efficient photocatalytic degradation of organic dyes were synthesized through gel combustion technique and characterized through X-ray Diffraction spectroscopy (XRD), UV-visible Spectroscopy, SEM, XPS, N2 adsorption-desorption, Thermo gravimetric, Ultra-Vis spectrophotometer and photocatalytic activity. But still, there are some major issues which need to address before the possible publication in this journal.

Minor revision.

Ø  In Abstract Author cut the description of experimental part and cut the line “Dye wastewater is one of the main sources of industrial wastewater, because of its com-8 plex composition, high chrominance, not easy biochemical degradation and other characteristics, 9 it is difficult to be effectively degraded by traditional water treatment technology. Photocatalysis 10 technology is a sustainable means to treat dye wastewater, it can combine photocatalyst with light 11 energy, has the advantages of high degradation efficiency, fast reaction speed, no secondary pollution and so on.

Ø  Author should discuss about XPS in introduction.

Ø  Introduction is to short author should improve the introduction, add some and may use references, https://doi.org/10.1016/j.ceramint.2020.12.234. https://doi.org/10.1007/s11356-022-19271-2 .

Ø  In XRD section Author should set the “Figure 2, Figure (a)”

Ø  Author should improve the XRD Section, find some structural parameters and may use references, https://doi.org/10.1007/s00339-023-06701-2.

Ø  In Uv-vis section Author says that “On the other hand, composites have large specific surface areas” explain it?

Ø  In Uv-Vis section author should explain it briefly and may use some references https://doi.org/10.1016/j.optmat.2020.110606.

Ø  What is the relation of BET with photo catalysis Author should give some reference

Ø  In photocatalytic section Author should perform a comparison table against MB dye.

Ø  In Fig.4 why author chose 5 cycle in cyclic stability of composites.

Ø  What is novelty of your work?

Comments on the Quality of English Language

  In this manuscript there are various grammatical inconsistencies throughout the manuscript so, Author needs to modify the grammar.

Author Response

Response to Reviewer 3 Comments

Point 1: In Abstract Author cut the description of experimental part and cut the line “Dye wastewater is one of the main sources of industrial wastewater, because of its com-8 plex composition, high chrominance, not easy biochemical degradation, and other characteristics, 9 it is difficult to be effectively degraded by traditional water treatment technology. Photocatalysis 10 technology is a sustainable means to treat dye wastewater, it can combine photocatalyst with light 11 energy, has the advantages of high degradation efficiency, fast reaction speed, no secondary pollution and so on.”

Response 1: The sentences in the above summary have been deleted and revised

Point 2:  Author should discuss about XPS in introduction.

Response 2: XPS is covered in lines 95 through 104 of this article

Point 3: Introduction is to short author should improve the introduction, add some and may use references,

Response 3: The introduction has been revised and relevant literature proposed by reviewers has been added. See the attached manuscript for details.

Point 4: In XRD section Author should set the “Figure 2, Figure (a)”

Response 4: The xrd picture is shown in Figure 1 below the analysis text. Figure 2 is the sem part.

Point 5: Author should improve the XRD Section, find some structural parameters and may use references,

Response 5: The xrd part has been modified, and the relevant literature proposed by the reviewer has been added for reference.

Point 6: In Uv-vis section Author says that “On the other hand, composites have large specific surface areas” explain it?

Response 6: Based on the experimental data in the BET test part of the following text, relevant conclusions are drawn, for details, see lines 129 to 153 of the manuscript.

Point 7: In Uv-Vis section author should explain it briefly and may use some references

Response 7: T In the UV-VIS part, I consulted and added the papers provided by the reviewers, and made modifications and explanations.

Point 8: What is the relation of BET with photo catalysis Author should give some reference

Response 8: BET surface area is an important property of porous materials, which is related to the pore structure and pore size of porous materials. A larger BET surface area means that more active surfaces are available for reactions to occur, providing more reaction sites and adsorption sites. This can increase the activity of the photocatalytic reaction because more reactants can meet the photocatalyst and react.

Point 9: In photocatalytic section Author should perform a comparison table against MB dye.

Response 9: In addition to methylene blue, the study of catalytic properties in other organic pollutants has been added, such as methyl orange, Rhodamine B, reactive brilliant red X-3B. It is found that the material also has a good ability to degrade these organic pollutants, and I have compared the performance of these results. We can see lines 237 to 251 of the manuscript.

Point 10: In Fig.4 why author chose 5 cycle in cyclic stability of composites.

Response 10: After repeated cycle experiments, we found that after the sixth cycle, the degradation rate of methylene blue was less than 80%, which could not achieve our expected effect, so it was abandoned in the paper.

Point11: What is novelty of your work?

Response11: 1. A new structure of three-dimensional CdZnS particles combined with metal-organic skeleton was constructed

  1. CdZnS composite not only broadens the absorption range of ZIF-8 to visible light, but also improves its photocatalytic activity
  2. A multistage porous composite material with microporous, mesoporous, and microporous structure was constructed.

Round 2

Reviewer 1 Report

Comments and Suggestions for Authors

The manuscript was improved accordingly.